# Expression of the Antimicrobial Peptide SE-33-A2P, a Modified Analog of Cathelicidin, and an Analysis of Its Properties

**DOI:** 10.3390/antibiotics13020190

**Published:** 2024-02-16

**Authors:** Vagif Gasanov, Ekaterina Vorotelyak, Andrey Vasiliev

**Affiliations:** Laboratory of Cell Biology, N.K. Koltzov Institute of Developmental Biology of Russian Academy of Sciences, Vavilov Str. 26, 119334 Moscow, Russia; vorotelyak@yandex.ru (E.V.); 113162@bk.ru (A.V.)

**Keywords:** cathelicidin, LL-37, antimicrobial peptide expression, AMP, SE-33, recombinant protein, cloning, A2P, cholesterol

## Abstract

In this study, we developed a method for the expression of the antimicrobial peptide SE-33-A2P in *E. coli* bacterial cells. The SE-33-A2P peptide consists of A2P and SE-33 peptides and is a retro analog of cathelicidin possessing antimicrobial activity against both Gram-positive and Gram-negative bacteria. Furthermore, the A2P peptide is a self-cleaving peptide. For an efficient expression of the SE-33-A2P peptide, a gene encoding several repetitive sequences of the SE-33 peptide separated by A2P sequences was created. The gene was cloned into a plasmid, with which *E. coli* cells were transformed. An induction of the product expression was carried out by IPTG after the cell culture gained high density. The inducible expression product, due to the properties of the A2P peptide, was cleaved in the cell into SE-33-A2P peptides. As the next step, the SE-33-A2P peptide was purified using filtration and chromatography. Its activity against both Gram-positive and Gram-negative bacteria, including antibiotic-resistant bacteria, was proved. The developed approach for obtaining a prokaryotic system for the expression of a highly active antimicrobial peptide expands the opportunities for producing antimicrobial peptides via industrial methods.

## 1. Introduction

Recently, mankind has faced the global problem of antimicrobial resistance [1,2]. This situation is particularly alarming in the case of multidrug-resistant Gram-negative bacterial pathogens [3,4,5] due to the fact that Gram-negative bacteria have an additional outer membrane surrounding their cells, which creates a permeability barrier for most hydrophobic molecules making them resistant to some antibiotics [6]. The negative charge of Gram-negative bacterial membranes is mainly due to lipid components. In Gram-negative bacteria, such as *Escherichia coli*, the main component of the outer membrane is a highly anionic glycolipid lipopolysaccharide (LPS) [7,8,9,10,11]. The negative charges of LPS are stabilized by divalent cations [12,13]. For *E. coli* strains, the highest degree of multidrug resistance has been shown [14]. Consequently, new strategies for killing Gram-negative bacteria without the rapid development of resistance have become a priority [15]. The solution to this problem is the development of new antimicrobial agents. Nevertheless, the number of annually registered antibiotics is decreasing [16] because the development of classical antimicrobial drugs is very costly, and the potential profit may not cover the costs due to the rapid development of resistance to these drugs [17]. From this point of view, it is necessary to look for compounds, to which bacteria will not be able to form resistance or will form resistance very slowly (over decades). In recent years, so-called antimicrobial peptides (AMPs), which are produced by all living species, have aroused great interest. They are often referred to as natural antibiotics [18,19], and resistance to AMPs develops much less frequently than to conventional antibiotics [20].

In addition to antibacterial action, many AMPs have antiviral, antifungal, antiparasitic, and antitumor effects [21]. They are natural peptides up to 100 amino acids long that have a great structural diversity [22,23,24]. Most mechanisms of AMPs’ action are based on the electrostatic binding of a positively charged peptide to the negatively charged components of the outer membrane of the bacterial cell. Binding to the membrane followed by the incorporation of the peptide into the lipid bilayer of the bacterial cell leads to pore formation [25,26], lipid bilayer chaotization [27], and even membrane collapse due to the resulting potential difference [28]. Each of these types of effects on a microorganism or their combination ultimately leads to the destruction of the microorganism. Therefore, one of the most important properties of AMPs necessary for their antibacterial properties to manifest is the ability to form an amphipathic alpha helix [29]. Particular attention has been drawn to cathelicidins, primarily to human cationic alpha-helical AMPs: peptide LL-37 and its precursor hCAP18 [30,31]. LL-37 (LLGDFFRKSKEKKEKIGKEFKKRIVQRIKDFLDFLRNLVPRTES), which has 37 amino acid residues, is produced by human mesenchymal stem cells, mucosal epithelial cells, neutrophils, monocytes, macrophages, mast cells, NK cells, T- and B-lymphocytes, adipocytes, and keratinocytes [32,33,34]. It is the only antibacterial peptide of the cathelicidin subfamily synthesized by human cells [35]. This peptide has activity against Gram-positive and Gram-negative bacteria, antiviral, antiparasitic, antifungal, antimalarial, and antitumor activity, participates in immune response modulation, and inhibits the activity of some enzymes [36,37,38,39].

A group of researchers [40] evaluated the ability of various compounds to form an amphipathic alpha helix (http://aps.unmc.edu/AP (accessed on 14 February 2024) and showed that one of the most promising candidates in this respect is the SE-33 peptide (SETRPVLNRLFDKIRQVIRKFEKGIKEKSKRFF), which has a retro-amino acid sequence compared to the natural human cathelicidin LL-37. They synthesized SE-33 via solid-phase synthesis and showed in in vitro experiments that the SE-33 polypeptide has a bactericidal effect on *Staphylococcus aureus* Wood 46 cells comparable with that of the LL-37 cathelicidin, as well as marked antifungal activity against clinical isolates of *Candida albicans*, *Cryptococcus neoformans*, *Rhodotorula mucilaginosa*, *Trichosporon cutaneum*, and *Geotrichum* sp.

To obtain a sufficient number of AMPs, three main approaches can be distinguished: direct isolation from natural sources, chemical synthesis or recombinant expression [41]. Since most organisms contain very low concentrations of AMPs, the isolation of peptides directly from natural sources is, in most cases, a labor-intensive and expensive process [42]. The chemical synthesis of peptides makes it possible to obtain both natural and synthetic AMPs, but a large-scale synthesis of peptides is expensive [43,44].

Advances in recombinant DNA technology have provided the ability to produce AMPs in large quantities [45]. This technology allows the cloning of foreign genes into specific vectors for expression in prokaryotic and/or eukaryotic host cells [46]. Expression in bacterial cells and yeasts is predominantly used, but expression in plants is also beginning to gain popularity [47,48,49].

Expression in cells is considered the most efficient method in terms of time and production costs [50]. It is possible to express AMPs without modification, but there is a threat of death of the producer cell under the influence of AMPs. Another way is to express an AMP fused to a carrier protein or express an AMP in the form of multimers, and the resulting product would need to be treated with specific proteases [51,52]. Due to the fact that chemical synthesis is quite expensive and has limitations for generating large amounts of peptides, we have attempted to express SE-33 in *E. coli* [53]. Since *E. coli* cells are not suitable for small peptides expressed in high concentrations, and AMP toxicity to the host cannot be ignored [54], we proposed an approach allowing us to express proteins with a molecular weight greater than 20 kDa consisting of the SE-33 peptide repeats, both native and modified, separated by methionine molecules. Such proteins had no toxic effect on *E. coli*. After expression and purification, these proteins were hydrolyzed with cyanobromide. As a result, the following peptides were obtained: the SE-33 peptide; the N6H-SE33 peptide, which constitutes the SE-33 peptide modified by a polyhistidine sequence (6 His) at the N-end; and the SE33-6HC peptide, which constitutes the SE-33 peptide modified by a polyhistidine sequence (6 His) at the C-end. All these peptides additionally had a methionine residue at the N-terminus. A study of their properties showed that all of them had antimicrobial properties against both Gram-positive and Gram-negative bacteria, including antibiotic-resistant bacteria. Thus, we were able to develop a method for obtaining active AMPs based on the SE-33 peptide using only two steps of chromatographic purification: affinity chromatography on Ni-NTA and, after hydrolysis with bromocyanine, ion-exchange chromatography on MonoQ.

However, the polyhistidine sequence disrupted the homogeneity of the amphipathic alpha helix of the SE-33 peptide, resulting in reduced antimicrobial properties of the N6H-SE33 and SE33-6HC peptides compared to SE-33. Our calculations showed that the addition of the amino acid motif EGRGSLLTCGDVEENPGP (A2-peptide) was not likely to significantly affect the homogeneity of the amphipathic alpha helix of the SE-33 peptide.

The A2P peptide is a self-cleaving peptide [55]. For an efficient expression of the SE-33-A2P peptide, a gene encoding several repetitive sequences of the SE-33 peptide separated by A2P sequences was created. During translation, A2P separates one SE-33 sequence from another, while A2P itself remains at the C-terminus of the previous SE-33 peptide [56,57]. The resulting SE-33-A2P peptide (SETRPVLNRLFRLFDKIRQVIRQVIRKFEKGIKEKSKRFFEGRGSLLTCGDVEENPGP) is intended to accumulate in the cell. The protein construct encoding five sequentially connected SE-33-A2P peptides was named rec5SE-33-A2P.

We have developed a method for the isolation of the SE-33-A2P peptide from pro-karyotic cells, which we characterized by physicochemical methods and susceptibility testing. This isolation method makes it possible to obtain SE-33-A2P AMP at a relatively low cost.

## 2. Results and Discussion

### 2.1. The Structure of the AMPs Used in the Study

Peptide SE-33, highly active against bacteria and fungi, is a retro analog of cathelicidin LL-37 (Figure 1). In addition to SE-33 itself, its synthetic analog and polyhistidine analogs were studied both at the N-end and at the C-end [52]. Based on the fact that the modified C-terminal analog is less active, it was decided to try to create the conditions for the expression of the active peptide directly in bacterial cells. Similar to earlier studies, we decided to use a repeated sequence of the SE-33 peptide separated by fragments that would have the ability to excise themselves. Of a series of A2 peptide modifications [58], A2P was chosen because its structure had a minimal effect on the alpha helix of the SE-33 peptide, and also A2P is the most active of all the peptides described in this series.

Given that protein synthesis is only possible from methionine and that we did not want to introduce additional amino acids into the SE-33 peptide sequence, it was decided to isolate methionine from the main array of repeating SE-33 peptides by the same A2P. Realizing that multiple repeats would only reduce the translation efficiency, we decided to leave five repeats of the SE-33 peptide in association with A2P (Figure 2).

### 2.2. Production of SE-33-A2P in E. coli

We expected that once the biomass reached high-density values and we induced IPTG expression, a large number of cells would begin to produce the rec5SE-33-A2P protein, which would be cleaved into SE-33 peptides during translation. Since some time was required for interaction with the membrane, it was expected to occur on the cytosolic side, and the activity of the derived peptide was not supposed to be high. We expected that cells would have time to produce some amount of the SE-33-A2P peptide before the start of cell death. Preparation of the pET5SEA2P plasmid (plasmid pET30a(+) was taken as the basis) carrying the gene encoding rec5SE-33-A2P and the transformation of *E.coli* BL21(DE3) with this plasmid caused no difficulties. However, when the transformed cells were growing in a medium with a selective antibiotic, a decrease in the rate of optical density of the culture was observed, and the maximum density of the cell culture transformed with the pET5SEA2P plasmid during daily incubation was almost three times lower than that of BL21(DE3) cells transformed with the pET30a(+) plasmid (Figure 3). This is probably due to the fact that there is little expression in the transformed plasmid. This, in turn, indicated that the SE-33-A2P peptide resulting from this background expression had antimicrobial properties with respect to the host cells.

Based on studies on the selection of optimal expression conditions in bacterial cells [59], we decided to add some components to the culture medium that reduce spontaneous expression. We hypothesized that spontaneous expression is triggered by trace amounts of lactose in the LB buffer (the native inducer of the promoter in the pET vector). The property of the pET vector is such that glucose serves as an inhibitor of the promoter induction, i.e., as long as sufficiently high concentrations of glucose are maintained in the cell, native inducers are not able to provoke expression. Glucose is often used to regulate autoinduction [60]. Moreover, glucose is one of the main energy sources in *E. coli*. However, if glycerol is also used as an energy source for the cell, it is metabolized first [61]. Thus, if using a mixture of glucose and glycerol in various proportions, it is possible to regulate not only autoinduction but also spontaneous expression.

Using LB mixtures with different concentrations of glucose and glycerol, spontaneous expression was completely abolished when using Mixture 3 (0.2% glucose and 0.15% glycerol), as seen in Figure 4. Therefore, further cultivation was carried out in the LB buffer containing 0.2% glucose and 0.15% glycerol in addition to kanamycin.

New difficulties arose during IPTG induction. An attempt to induce when the maximum optical density of the cell culture (OD560nm = 3.6) was reached resulted in the formation of a dense foam and an almost fivefold decrease in optical density (to a value of OD560nm = 0.69), indicating a total lysis of the bacterial cells. A similar result was obtained upon induction of the expression when the optical density of the cell culture reached a value of OD560nm = 2.0. The foam also appeared within 10–15 min, and after 30 min, there was a decrease in optical density to a value of OD560nm = 0.52 (more than fourfold), which also indicated the lysis of the bacterial cells.

Based on the culture studies during cultivation [62], the optimal concentration of the cell culture for induction (OD560 = 2.2) was selected. And, based on the studies showing the positive effect of cholesterol on the cell membrane [63], we decided to use it to restore membranes. Several cholesterol solutions were prepared (Chol 0 mM, Chol 5 mM, Chol 25 mM, Chol 50 mM and Chol 100 mM) and added to the culture medium when its optical density reached OD560nm = 0.5, and IPTG induction was performed when the optical density reached OD560nm = 2.2 (Figure 5).

### 2.3. Extraction and Purification of SE-33-A2P and Evaluation of Its Physicochemical Properties

The culture medium with 50 mM cholesterol was lysed 45 min after IPTG induction. After precipitation via centrifugation, the bacterial lysate was applied to centrifuge columns/tubes equipped with 100, 30, and 10 kDa filter membranes. After each new filtration, the lysate was stripped of large molecules. After the last filtration, the lysate was chromatographed on Mono Q (Figure 6).

Fractions 1–6 of the Mono Q chromatogram were selected for evaluation of the peptide presence via Tricine-SDS-PAGE (Figure 7). The purest fraction was subjected to MALDI-TOF analysis (Figure 8) and analytical HPLC (Figure 9). The quality of the resulting peptide was very high. In Figure 8, the major peak has a mass of 5909.5 Da, which corresponds to the theoretical molecular mass of 5907.81 Da for the SE-33-A2P peptide. Analytical HPLC showed the retention time of the major peak (32.4 min) corresponding to the yield of the SE-33-A2P peptide. The table in Figure 9 reflects the purity of the peptide as a percentage of the total number of peptides. The purity of the SE-33-A2P peptide was 92.8%, which was confirmed via HPLC.

Nevertheless, the fractions containing the target product were combined, diluted 20-fold with the buffer, with which the Mono Q column was equilibrated, and reapplied to Mono Q (Figure 10). The fractions were again analyzed via MALDI-TOF (Figure 11) and HPLC (Figure 12).

There is only one peak in Figure 11, and it has a mass of 5909.5 Da, which corresponds to the calculated mass of 5907.8 Da for the SE-33-A2P peptide. The analytical HPLC retention time of the peak corresponding to the yield of the SE-33-A2P peptide was 32.4 min. In Figure 12, there is only one peak, which indicates the high purity of the SE-33-A2P peptide. The purity of the SE-33-A2P peptide was close to 100% then, which was confirmed via mass spectrometry and HPLC.

Cultivating BL21(DE3) transformed with the pET5SE33A2P plasmid in 1 L of medium with Buffer 3 (LB + 0.2% glucose and 0.15% glycerol) and Chol Buffer 50 mM (LB + 50 mM cholesterol) with two chromatographic purifications on Mono Q yielded 56 mg of the purified SE-33-A2P peptide.

### 2.4. Evaluation of the Antimicrobial Properties of SE-33-A2P

The antimicrobial properties of the peptides were investigated on the Gram-negative bacteria of the seven strains of *E. coli*: (BL21(DE3), BL21(DE3)pLysS, BLR(DE3), Rosetta2 (DE3), Rosetta2(DE3)pLysS, Origami(DE3), and Origami(DE3)pLysS), including those with antibiotic resistance, as well as on the Gram-positive bacteria (*Staphylococcus aureus* and *Lactobacillus casei* (ApR+ and ApR−). Peptides SE-33-A2P and SE-33 in different concentrations were added to the liquid nutrient media with the bacteria. The antimicrobial properties of the SE-33-A2P peptide were assessed by measuring the concentration of the antimicrobial peptides SE-33 and SE-33-A2P entailing complete lysis of bacterial cells (minimum bactericidal concentration (MBC) [64]) after 24 h incubation and inoculation onto dishes with solid nutrient media (Table 1).

Interestingly, the antimicrobial activity of the SE-33-A2P peptide was only half that of the SE-33 peptide. The average MBC was 45 μg/mL for Gram-negative bacteria and 35 μg/mL for Gram-positive bacteria (Table 1).

The antimicrobial properties of the SE-33-A2P and SE-33 peptides were also evaluated by analyzing the antimicrobial effect on the total flora obtained from mouse excreta. The evaluation was performed by counting the number of colonies (Figure 13). The complete absence of bacterial growth on a Petri dish was achieved at the following concentrations: 20 μg/mL for the SE-33 peptide and 40 μg/mL for the SE-33-A2P peptide. The heterogeneity of the colonies on the plates represents the entire bacterial flora that is found in mouse excrement.

Thus, the presence of the A2P peptide at the C-terminus of the SE-33 peptide did not actually decrease its antimicrobial properties.

## 3. Materials and Methods

### 3.1. Escherichia coli Strains and Plasmids

*Escherichia coli* Strains and Plasmids that were used in the study a listed in Table 2.

### 3.2. Media to Reduce Spontaneous Expression

We used Buffer Chol 0 mM (LB [#X968.1 Carl Roth. Karlsruhe, Germany] + 0 mM cholesterol); Buffer Chol 5 mM (LB + 5 mM cholesterol [#C8667 Sigma-Aldrich. Saint Louis, MO, USA]); Buffer Chol 25 mM (LB + 25 mM cholesterol); Buffer Chol 50 mM (LB + 50 mM cholesterol); and Buffer Chol 100 mM (LB + 100 mM cholesterol).

### 3.3. Media to Stabilize Bacterial Membranes

We used Mixture 1 (LB + 0.5% glucose [#G7021 Sigma-Aldrich. Saint Louis, MO, USA]); Mixture 2 (LB + 0.2% glucose, 0.15% glycerol [#1613407 Sigma-Aldrich. Saint Louis, MO, USA], KCl 50 mM, MgCl_2_ 10 mM); Mixture 3 (LB + 0.2% glucose, 0.15% glycerol); and Mixture 4 (LB + 0.3% glycerol).

### 3.4. Cloning rec5SE-33-A2P, Obtaining the pET5SE33A2P Plasmid, and Obtaining the rec5SE-33-A2P Progenitor Strain

The DNA fragment encoding rec5SE-33-A2P was synthesized via Evrogen (Moscow, Russian Federation) and included sequences encoding the start amino acid sequence MEGRGSLLTCGDVEENPGP, which is methionine restricted by A2P, and 5 repeats of the SE-33-A2P peptide (SETRPVLNRLFDKIRQVIRKFEKGIKEKSKRFFGGSLLTCGDVEENPGP). The sequence was flanked by the BamHI [#ER0581 Thermo Fischer. Waltham, MA, USA] and HindIII [#ER0501 Thermo Fischer. Waltham, MA, USA] restriction endonuclease sites. The gene was optimized for expression in *E. coli* using the JCat software 3.04 (http://www.jcat.de/). The secondary structure of the rec5SE-33-A2P mRNA was optimized using the DINAMelt web service (http://mfold.rna.albany.edu/?q=DINAMelt/Twostatefolding (accessed on 5 February 2024)). The synthetic gene was cloned into the pET30a(+) plasmid [#69909 Sigma-Aldrich. Milwaukee, WI, USA] at the BamHI and HindIII sites. The resulting plasmid pET5SE33A2P was used to transform *E. coli* XL1-Blue cells.

The pET5SE33A2P plasmid was produced by growing *E. coli* XL1-Blue cells transfected with the pET5SE33A2P plasmid in the LB medium [#X968.1 Carl Roth. Karlsruhe, Germany] with selective antibiotic kanamycin at a concentration of 25 µg/mL [#A1493,0005 AppliChem. Milwaukee, WI, USA] to optical density values of OD560 nm = 0.6. Next, the pET5SE33A2P plasmid was isolated using a Plasmid Miniprep kit [#BC021S Evrogen. Moscow, Russian Federation] according to the manufacturer’s protocol (https://evrogen.ru/products/isolation/plasmid-kits).

*E. coli* BL21(DE3) cells were transfected with the isolated plasmid pET5SE33A2P. Competent *E. coli* BL21(DE3) cells were thawed in a water bath, 1.5 µL of plasmid solution was added, the mixture was heated to 42 °C for 2 min, then 1 mL of the LB nutrient medium was added, and the mixture was incubated for 1 h at 37 °C. From the culture broth, 50–100 µL was taken and seeded into 2 mL of the LB medium with the addition of the antibiotic kanamycin at a concentration of 25 µg/mL and cultured at 200 rpm and 37 °C in an incubator shaker (ES-20/60, P-6/1000, BioSan. Riga, Latvia) for 4–5 h to optical density values of OD560nm = 0.6. The culture medium was then seeded onto Petri dishes. For this purpose, 17 mL of agar (1.5% agar [#A7002 Sigma-Aldrich. Milwaukee, WI, USA] in the LB medium) was poured into 90 mm Petri dishes. After the agar solidified, 100 µL of kanamycin antibiotic solution was rubbed into it with a Drigalski spatula to achieve a final antibiotic concentration of 25 µg/mL in the agar. Next, 25 μL of the culture with the grown strain was added to the cup. This volume was rubbed into the agar with a Drigalski spatula, also. The mixture was left for 6–8 h at 37 °C. The grown colonies were used as a producer strain.

Optical density was assessed using NanoDrop (IMPLEN. Westlake Village, CA, USA).

### 3.5. Biomass Production and Expression Induction

A colony of the producer strain was inoculated in 5 mL of the LB medium supplemented with the antibiotic kanamycin at a concentration of 25 µg/mL. The cultivation was performed until the optical density reached OD560nm = 0.6. Next, glycerol was added to the solution to a final concentration of 15%, and 100 μL of the solution with the bacteria was packed into cryotubes. The producer strain stock was stored at −80 °C.

Bacteria biomass was grown in a 750 mL round-bottom culture flask at 220 rpm in the incubator shaker. Furthermore, 2 mL of the nutrient medium (Mixture 3 with 25 μg/mL kanamycin) and 100 μL of the producer strain stock, extracted from storage and thawed, were introduced into the flask. The mixture was incubated at 37 °C in the incubator shaker for 3 h. Next, 18 mL of nutrient medium (Mixture 3 and 25 μg/mL kanamycin) was added to the flask. The mixture was again incubated in the incubator shaker for 3 h. Then, 180 mL of nutrient medium (Mixture 3 and 25 µg/mL kanamycin) was added to the flask. The mixture was cultivated in the incubator shaker until the optical density reached OD560 nm = 2.2. Finally, the IPTG solution was added to the culture medium to a concentration of 0.5 mM.

### 3.6. Biomass Destruction

The cells were collected via centrifugation at 5000× *g* for 30 min at 10 °C and frozen at –20 °C. The thawed *E. coli* cells were mixed with 50 mL of a solution containing 20 mM TrisHCl (pH 8.5) [#C1046 Sigma-Aldrich. Milwaukee, WI, USA], 1 mM phenylmethylsulfonyl fluoride (PMSF) [#PMSF-RO Roche. Basel, Switzerland], 100 μg/mL lysozyme [#L2879 Sigma-Aldrich. Milwaukee, WI, USA], and 20 μg/mL DNase [#D5319 Sigma-Aldrich. Milwaukee, WI, USA]. After 5 min, a suspension of urea [#U5128 Sigma-Aldrich. Milwaukee, WI, USA] was added to a concentration of 6 M. The solution was incubated for 20 min at room temperature.

### 3.7. Purification of the Peptide by Filtration

The biomass lysate was centrifugated at 80,000× *g* for 45 min. The precipitate was removed. The supernatant was placed into a centrifuge filter column/tube [Spin-X, 100,000 MWCO, #431491, Corning. New York, NY, USA] with a 100 kDa cutoff. After centrifugation at 5000× *g* for 30 min, the filtrate was transferred to a centrifuge filter column/tube [Spin-X, 30,000 MWCO, #431489, Corning. New York, NY, USA] with a 30 kDa cutoff. After centrifugation at 5000× *g* for 30 min, the filtrate was transferred to a centrifuge filter column/tube [Spin-X, 10,000 MWCO, #431488, Corning. New York, NY, USA] with a 10 kDa cutoff. After centrifugation at 5000× *g* for 30 min, the filtrate was collected and used for chromatographic purification.

### 3.8. Purification of the Peptide by Chromatography

A Mono Q 4.6/100 PE column [#17517901, Cytiva. Stockholm, Sweden] was mounted on the AKTA Prime Plus (GE Healthcare. Chicago, IL, USA) and equilibrated with 20 mM TrisHCl (pH 8.5) and 5 mM NaCl. The entire volume of the lysate was applied. The lysate was eluted with a linear gradient. The start buffer consisted of 20 mM TrisHCl (pH 8.5) and 5 mM NaCl, and the finish buffer consisted of 20 mM TrisHCl (pH 8.5) and 120 mM NaCl.

The target product came out in the range of NaCl concentrations from 60 mM to 85 mM.

### 3.9. Analytical HPLC

The purity of the peptide preparation, SE-33-A2P, was assessed on a Kromasil C-18 HPLC column. The samples were preacidified to 0.2% TFA [#1.00565 Sigma-Aldrich. Milwaukee, WI, USA]. The chromatography was performed in a linear gradient of acetonitrile [#574732 Sigma-Aldrich. Milwaukee, WI, USA] (the start buffer: 0.2% TFA and 5% acetonitrile; the finish buffer: 0.2% TFA and 80% acetonitrile). The ratio of the peak areas corresponds to the ratio of the amount of the substance.

### 3.10. Tricine-SDS-PAGE

The separation was performed on the Mini Protean Tetra Cell (Bio-Rad. Hercules, CA, USA). The concentrating gel was 6% acrylamide, and the separating gel was 16.5% acrylamide. The gel composition was chosen according to [65,66]. Phoresis was performed at 30 mA for 65 min. The gel was stained with Coomassie Blue G-250 [Am-0615-5.0 VWR-Ameresco. Framingham, MA, USA] solution in 10% ethanol.

Plasmid DNA transformation, DNA cleavage, ligation, electrophoresis in agarose gels, and other procedures were performed using generally accepted methods [67].

### 3.11. MALDI-TOF

MALDI-TOF mass spectrometry was performed at the Department of Protein Chemistry of the Belozersky Institute of Physicochemical Biology of Lomonosov Moscow State University on the UltrafleXtreme mass spectrometer (Bruker. Bremen, Germany).

Molecular masses and other properties of recombinant polypeptides were calculated using the ExPASy web portal (http://web.expasy.org/cgibin/protparam/protparam (accessed on 5 February 2024)).

### 3.12. Evaluation of the Antimicrobial Properties of Peptides on Various Bacterial Strains

Bacterial cells of different *E. coli* strains were grown in the LB medium: XL-1Blue; BL21(DE3); AD494; BLR(DE3); BL21(DE3)pLysS; Origami(DE3); and Origami(DE3)pLysS. The cultivation was performed in the incubator shaker at 200 rpm at 37 °C in 250 mL culture flasks (the volume of the LB medium with bacterial cells placed into the flasks was 50 mL).

The antibiotic kanamycin was added to the LB medium where strain AD494 was grown; the final concentration of the antibiotic was 25 µg/mL. The antibiotic tetracycline was added to the LB medium where strain BLR(DE3) was grown at a final concentration of 12.5 µg/mL. The antibiotic chloramphenicol was added to the LB medium where strain BL21(DE3)pLysS was grown at a final concentration of 25 µg/mL. The antibiotic tetracycline at a final concentration of 12.5 µg/mL and the antibiotic kanamycin at a final concentration of 25 µg/mL were added to the LB medium where strain Origami(DE3) was grown. We added the following to the LB medium where strain Origami(DE3)pLysS was grown: the antibiotic tetracycline at a final concentration of 12.5 µg/mL, the antibiotic kanamycin at a final concentration of 25 µg/mL, and the antibiotic chloramphenicol at a final concentration of 25 µg/mL.

Bacterial cells of *Staphylococcus aureus* and *Lactobacillus casei* were also grown in the LB medium without adding antibiotics. *Lactobacillus casei* * bacterial cells that are resistant to ampicillin (created in our laboratory) were grown in the LB medium with the addition of the antibiotic ampicillin at a final concentration of 50 µg/mL.

All cells were grown in the media until the cell culture reached an optical density value equal to OD560nm = 0.6. Next, the SE-33-A2P and SE-33 AMPs proteins were added to the culture media with the bacteria at a final concentration of 0.1 µg/mL, 1 µg/mL, 5 µg/mL, 10 µg/mL, 15 µg/mL, 20 µg/mL, 25 µg/mL, 30 µg/mL, 35 µg/mL, 40 µg/mL, 45 µg/mL, 50 µg/mL, 55 µg/mL, 60 µg/mL, 65 µg/mL, 70 µg/mL, 75 µg/mL, 80 µg/mL, 85 µg/mL, 90 µg/mL, 95 µg/mL, and 100 µg/mL. After the addition of the AMPs, the culture media were incubated for 30 min in the incubator shaker at 50 rpm at 37 °C. Next, 100 μL of the culture medium was introduced into a Petri dish with 1.5% LB agar (without antibiotics) and was carefully distributed over the entire surface of the Petri dish using a Drigalski spatula. Three Petri dishes were used for each dilution of the AMP for different strains. After seeding, the Petri dishes were placed into a 37 °C thermostat. After 24 h, the number of colonies grown on the Petri dishes was evaluated; the minimum bactericidal concentration (MBC) was the minimum concentration of the AMP at which not a single colony grew on any of the three petri dishes.

Cultures of the same bacterial strains without incubation with the AMP were used as a negative control.

### 3.13. Evaluation of the Antimicrobial Properties of the Peptides by Seeding on Mouse Feces

The feces of a C57BL/6 strain male mouse weighing 25 g were used for the study. The standards for animal housing were those defined in Directive 2010/63/EU for the protection of animals used for scientific purposes. There was no litter contamination that could affect the results of the study. The animal was fed standard SSNIFF^®^ pellet feed (V1534-3 10 mm pellets, autoclavable), which was fed ad libitum into the feeding recess of the steel lattice cage cover.

No manipulation on the animal was performed as a result of the studies, and only excreta were used in the study.

Moreover, 100 mg of mouse excreta was resuspended in 5 mL of PBS buffer and incubated for 10 min at room temperature. Furthermore, 1 mL of the suspension was added to 50 mL of the LB buffer, and the total flora was grown in 250 mL culture flasks. The cultivation was performed in the incubator shaker at 37 °C at 200 rpm until the cell culture reached an optical density of 0.6 (560 nm). Next, the SE-33-A2P, SE33-6HC, and SE-33 AMPs were added to the culture media with the bacteria at the final concentrations described in step 2.13. After adding the AMPs, the culture media were incubated for 60 min in the incubator shaker at 50 rpm at 37 °C. Moreover, 100 μL of the culture medium was introduced into each of a series of Petri dishes with 1.5% LB agar (without antibiotics) and was carefully distributed over the entire surface of each Petri dish with a Drigalski spatula. Three Petri dishes were used for each dilution of AMPs in different strains. After seeding, the Petri dishes were placed into a 37 °C thermostat. After 48 h, the number of colonies on the Petri dishes was estimated. The minimum concentration of the AMP at which not a single colony grew on any of the three Petri dishes was taken as the minimum bactericidal concentration (MBC).

Cultures without incubation with AMPs were used as a negative control.

## 4. Conclusions

Peptide SE-33, a retro analog of cathelicidin, proved to be an active and stable AMP with a wide spectrum of action [51]. Different modifications of the SE-33 peptide were developed, and approaches to SE-33 production via biotechnological methods are being established [52]. We have developed a technological approach for the production of the modified analog of cathelicidin, the SE-33-A2P peptide, in prokaryotic cells and its purification using routine methods. This AMP possesses a pronounced antimicrobial action against Gram-positive and Gram-negative bacteria, including those resistant to antibiotics. The approach allows us to obtain an AMP purity over 92% by performing minimal manipulations. Moreover, the purity of the AMP increases to 100% via repeated chromatography. The use of self-cleaving A2P peptides in the production of the SE-33-A2P peptide suggests that their use is potentially promising for obtaining other AMPs. Both the high activity of the SE-33-A2P peptide and the high-tech method for its production make this AMP very attractive for further research.

## Figures and Tables

**Figure 1 antibiotics-13-00190-f001:**
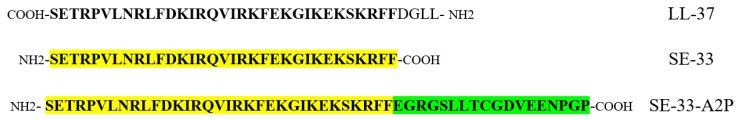
Amino acid composition of antimicrobial peptides. The amino acid sequence of the SE-33 peptide is highlighted in yellow. The amino acid sequence of the A2P peptide is highlighted in green.

**Figure 2 antibiotics-13-00190-f002:**
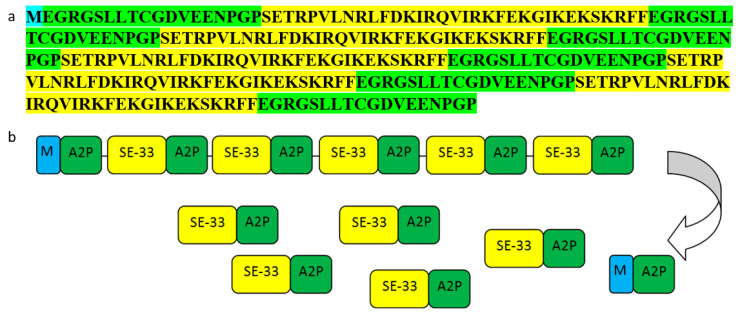
Amino acid composition (**a**) and the structure of the rec5SE-33-A2P protein and its products after hydrolysis (**b**). The amino acid sequence of the SE-33 peptide is highlighted in yellow. The amino acid sequence of the A2P peptide is highlighted in green. The amino acid methionine is highlighted in blue.

**Figure 3 antibiotics-13-00190-f003:**
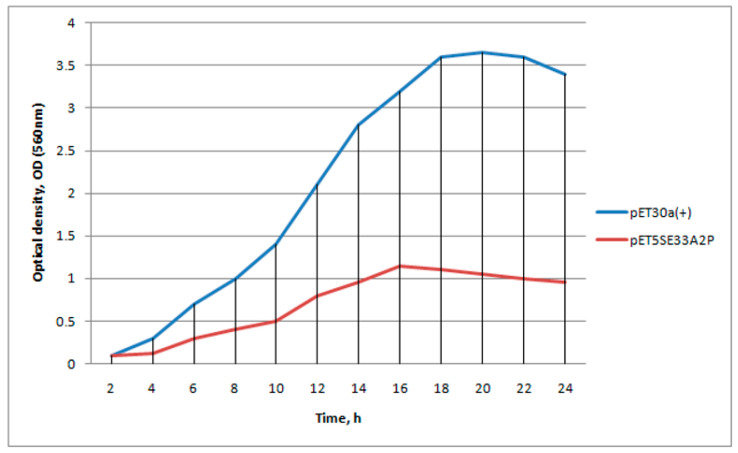
Optical density changes in the BL21(DE3) cell cultures transformed with the pET30a(+) and pET5SE33A2P plasmids.

**Figure 4 antibiotics-13-00190-f004:**
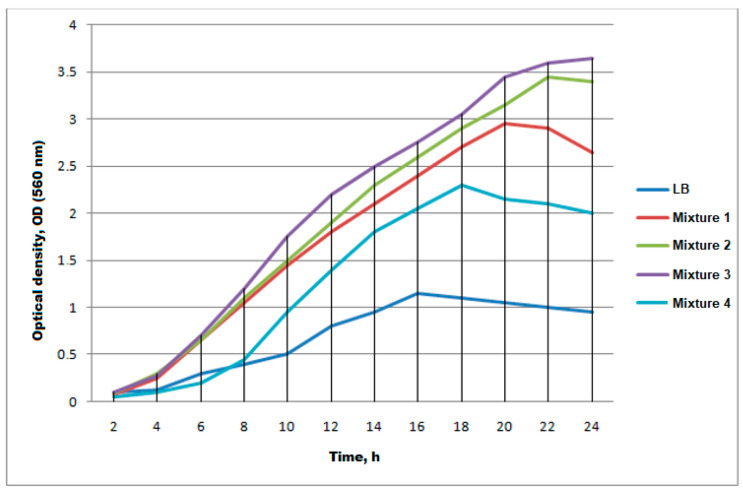
Changes in the optical density of the BL21(DE3) cell culture transformed with the pET5SE33A2P plasmid in the presence of different mixtures.

**Figure 5 antibiotics-13-00190-f005:**
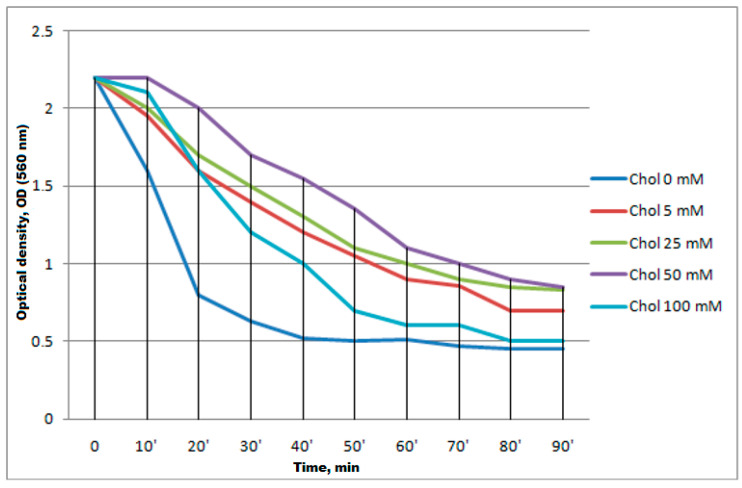
Changes in the optical density of the BL21(DE3) cell culture transformed with the pET5SE33A2P plasmid after induction depending on the concentration of cholesterol in the culture medium.

**Figure 6 antibiotics-13-00190-f006:**
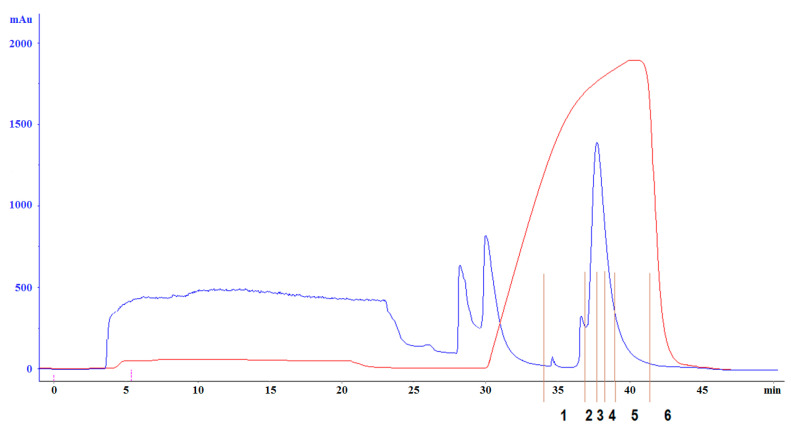
SE-33-A2P peptide chromatography on Mono Q. The blue line corresponds to the change in optical density, and the red line corresponds to the change in the salt composition of the buffer. Vertical orange lines and black numbers indicate which section of the chromatogram corresponds to fractions 1–6.

**Figure 7 antibiotics-13-00190-f007:**
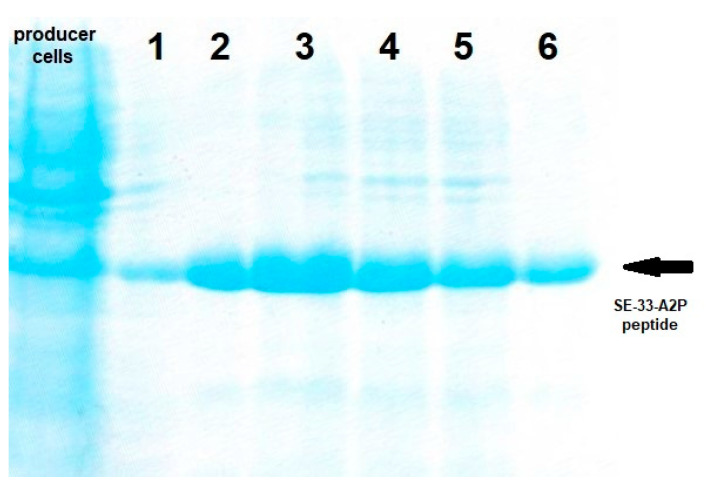
Tricine-SDS-PAGE (16.5%) electrophoresis of the BL21(DE3) cell culture transformed with the pET5SE33A2P plasmid expressing the SE-33-A2P peptide (producer cells), as well as fractions 1–6 (marked with numbers 1–6); the arrow marks the SE-33-A2P peptide.

**Figure 8 antibiotics-13-00190-f008:**
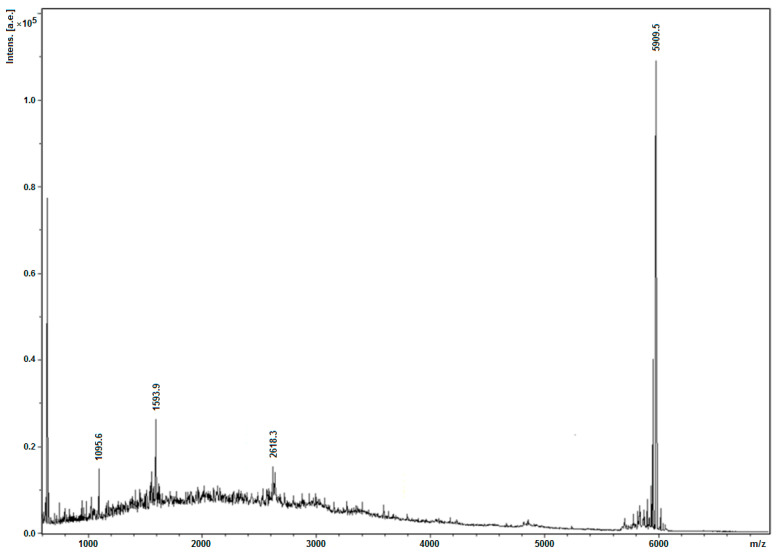
MALDI-TOF of the SE-33-A2P peptide after the first chromatography on Mono Q: the resulting mass of the SE-33-A2P peptide is 5909.5 Da (while the estimated mass of the SE-33-A2P peptide is 5907.8 Da).

**Figure 9 antibiotics-13-00190-f009:**
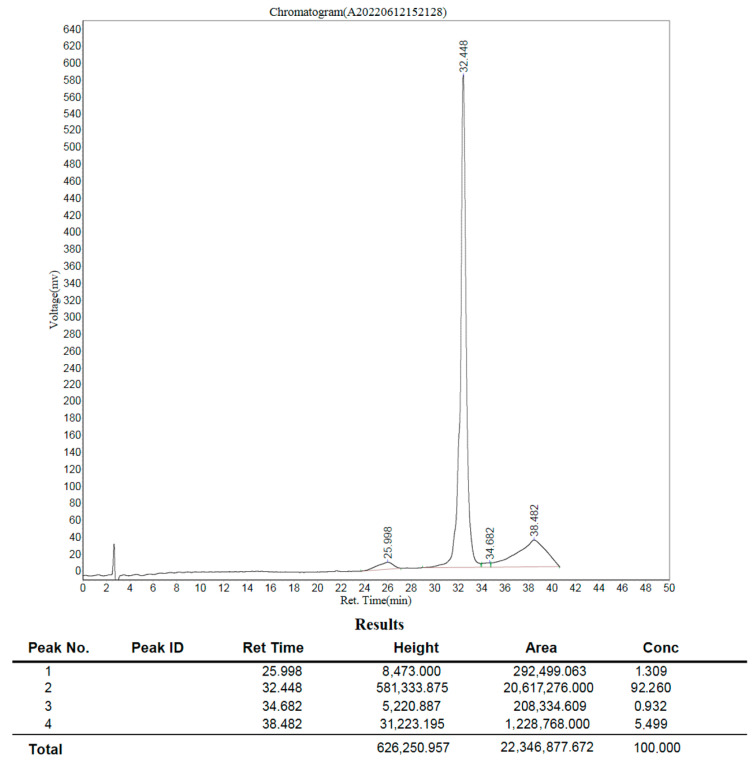
Analytical HPLC of the SE-33-A2P peptide after the first chromatography on MonoQ: peptide purity is 92.8%, retention time is 32.4 min.

**Figure 10 antibiotics-13-00190-f010:**
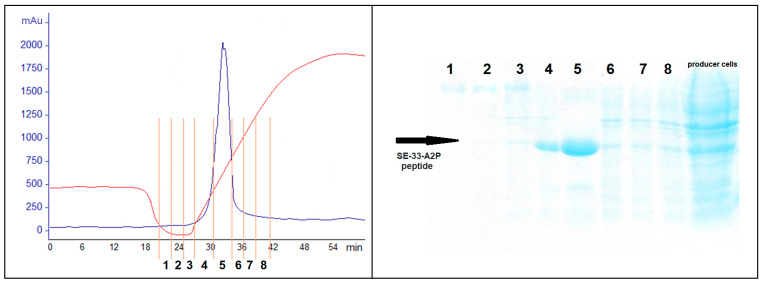
Repeat chromatography of the SE-33-A2P peptide on MonoQ (on the left). The blue line corresponds to the change in the optical density, the red line corresponds to the change in the salt composition of the buffer, and vertical orange lines and black numbers indicate which section of the rechromatogram corresponds to fractions 1–8. Tricine-SDS-PAGE (16.5%) electrophoresis of the BL21(DE3) cell culture transformed with the pET5SE33A2P plasmid expressing the SE-33-A2P peptide (producer cells), as well as fractions 1–8 (marked with numbers 1–8) (on the right); the arrow marks the SE-33-A2P peptide.

**Figure 11 antibiotics-13-00190-f011:**
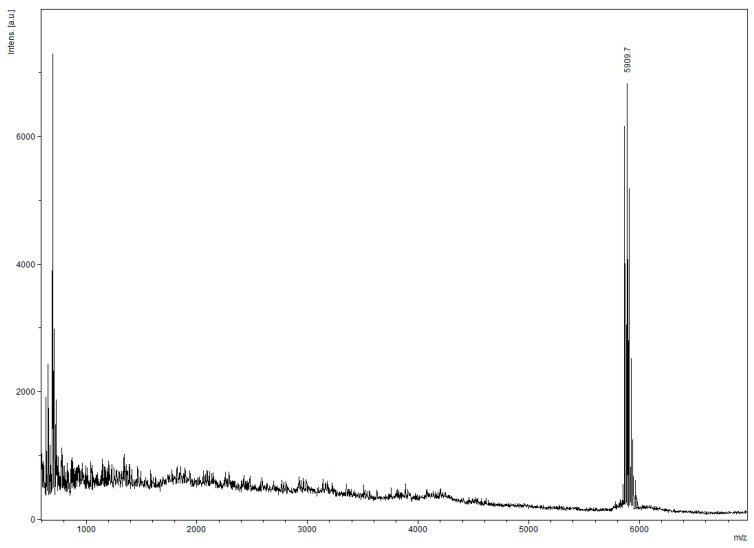
MALDI-TOF of the SE-33-A2P peptide after repeated chromatography on Mono Q showing the resulting mass of the SE-33-A2P peptide 5909.7 Da.

**Figure 12 antibiotics-13-00190-f012:**
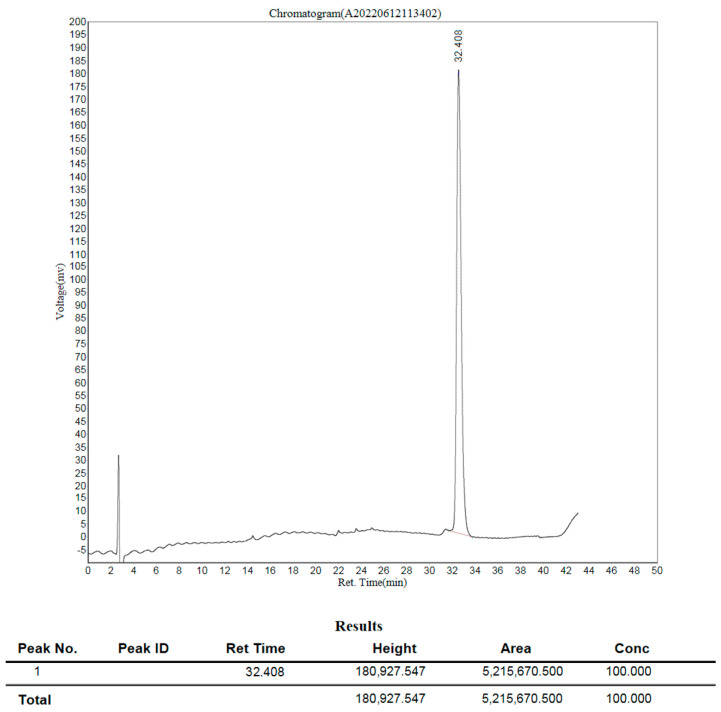
Analytical HPLC of the SE-33-A2P peptide after rechromatography on MonoQ showing 100% peptide purity and 32.4 min retention time.

**Figure 13 antibiotics-13-00190-f013:**
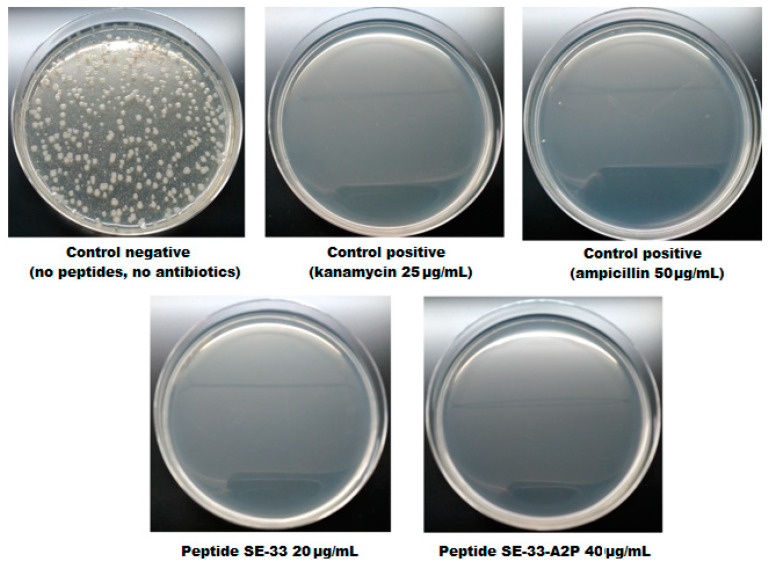
Evaluation of the antimicrobial properties of the SE-33 and SE-33-A2P peptides via inoculation of mouse feces on Petri dishes.

**Table 1 antibiotics-13-00190-t001:** Concentrations of antimicrobial peptides SE-33 and SE-33-A2P entailing complete lysis of bacterial cells.

Bacterial Cells	Gram-Negative							Gram-Positive		
	XL-1	BL21(DE3)	AD494	BLR(DE3)	BL21(DE3)pLys S	Origami(DE3)	Origami (DE3)pLysS	*Staphylococcus aureus*	*Lactobacillus casei*	*Lactobacillus casei **
Resistance	No	No	Km^R^	Tc^R^	Cm^R^	Km^R^ Tc^R^	Km^R^ Cm^R^ Tc^R^	No	No	Ap^R^
	Minimum Bactericidal Concentration (MBC), µg/mL									
Peptide SE-33	15	15	20	20	20	20	20	15	20	20
Peptide SE-33-A2P	35	35	45	40	45	45	45	35	30	35

No, no antibiotic resistance; Km^R^, resistance to kanamycin; Cm^R^, resistance to chloramphenicol; Tc^R^, tetracycline resistance; Ap^R^, ampicillin resistance; *, modified Lactobacillus casei, which is resistant to ampicillin.

**Table 2 antibiotics-13-00190-t002:** *Escherichia coli* Strains and Plasmids.

Name of *E. coli* Strain and Plasmids	Properties	Developer and Catalog Number
*E. coli* XL-1 Blue	*E. coli* non-T7, recA, endA, B/W scr, lacl, F’episome, lacY, Available as Singles, Available as HT96, Tetr R	Novagen, #69825 Sigma-Aldrich. Milwaukee, WI, USA
*E. coli* BL21(DE3)	*E. coli* pET, ompT, lon, dcm, Available as Singles, Available as HT96	Novagen, #69450 Sigma-Aldrich. Milwaukee, WI, USA
*E. coli* AD494	*E. coli* non-T7, lacl, F’episome, trxB, Km R	Novagen, #69845 Sigma-Aldrich. Milwaukee, WI, USA
*E. coli* BLR(DE3)	*E. coli* pET, recA, ompT, lon, dcm, Tetr R	Novagen, #69053 Sigma-Aldrich. Milwaukee, WI, USA
*E. coli* BL21(DE3)pLysS	*E. coli* pET, ompT, lon, pLysS, dcm, Available as Singles, Chl R	Novagen, #69451 Sigma-Aldrich. Milwaukee, WI, USA
*E. coli* Origami(DE3)pLysS	*E. coli* pET, lacl, F’episome, trxB, gor, Available as Singles, Km R, Tetr R, Chl R	Novagen, #71431 Sigma-Aldrich. Milwaukee, WI, USA
*E. coli* Origami	*E. coli* non-T7, lacl, F’episome, trxB, gor, Km R, Tetr R	Novagen, #71345 Sigma-Aldrich. Milwaukee, WI, USA
pET30a(+)	Plasmid	Novagen, #69909 Sigma-Aldrich. Milwaukee, WI, USA

## Data Availability

The data presented in this study are available in https://drive.google.com/drive/folders/10nyhRT0902ZxSntVijePWDobgCB1Atbi?usp=sharing.

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
