# Peer review of "Expression of the Antimicrobial Peptide SE-33-A2P, a Modified Analog of Cathelicidin, and an Analysis of Its Properties"

_antibiotics, 2024, doi:10.3390/antibiotics13020190_

Round 1
Reviewer 1 Report
Comments and Suggestions for Authors
The authors have provided a thorough characterization of the analogs of Cathelicidin produced in the study. Some of the sentence constructions are a little off, but there was no difficulty in understanding the study. The authors provided an in depth analysis of the products made. The approach to express AMPs in a bacterial expression host is interesting.
Comments on the Quality of English LanguageThe English is understandable
Author Response
Dear Reviewer!
Thank you very much for your time, evaluation of our manuscript and giving a high score. However, other reviewers made some remarks, and we had to make changes to the manuscript. Please, refer to the new submitted file.
Reviewer 2 Report
Comments and Suggestions for Authors
The authors constructed a vector to express multiple SE-33 AMP fragments separated by 2A peptides, which yields functional AMPs but with slightly lower efficacy. This paper is straightforward, easy to understand and easy to practice. There are some minor concerns. This strategy includes a 2A sequence followed by the SE-33, which means a “new” AMP is developed. Therefore, it is necessary to compare to the original SE-33 not only to the antimicrobial efficacy, but also include toxicity etc.
Small corrections:
1. “E. coli” and other strains should be italic.
2. There are many sentences should be rewritten to make the idea more clear. E.g. “Preparation of the pET5SEA2P plasmid (plasmid pET30a(+) was taken as the 164 basis) carrying the gene encoding rec5SE-33-A2P and transformation of BL21(DE3) E.coli 165 with this plasmid caused no difficulties.”
3. As a strategy for heterologous expression of AMPs, the authors did not do a comparative survey of how other strategies can yield, which is a pity, such as Peptides, Volume 38, Issue 2, December 2012, Pages 446-456, https://doi.org/10.1016/j.peptides.2012.09.020; Natural Product Reports 2023, DOI: 10.1039/D3NP00031A, http://dx.doi.org/10.1039/D3NP00031A; Protein Expression and Purification, Volume 140, December 2017, Pages 52-59, https://doi.org/10.1016/j.pep.2017.08.003
4. Result table 2/figure 4. The option of different buffers, I would like the authors include the literatures to support the choices. I would also like the authors to explain the differences caused by the different buffers.
5. Figure 6. Please give a short description of the figure curves.
6. Figure 7. Why did the authors not include protein maker on the SDS-PAGE? Also in figure 10.
7. Table 4. Are the concentrations of SE-33-A2P determined under normal culture medium, since authors used different combinations of medium in expression tests.
8. The sentence of the result “EC100 averaged 0.45 μg/ml for Gram-negative bacteria and 0.35 for 259 Gram-positive bacteria (Table 4)”. It is confusing since in the table 4, the EC100 should be 15-45µg/mL.
9. Figure 13. The colonies on the control plate seems no pure enough. It does not impair the conclusion, but it would be much accurate to know which pathogens of the AMP can kill. Additionally, there is no control of conventional antibiotics in the antimicrobial testing.
Comments on the Quality of English Languagesee comments
Author Response
Dear Reviewer! Thank you very much for your review. We responded to all comments, corrections have been made. We would like to draw your attention to the fact that due to the presence of comments from other experts, the revised text also contains additional edits.

Reviewer 3 Report
Comments and Suggestions for Authors
Major.
The introduction needs to be shortened greatly before consideration for publication. Is it introduction or results from a previous publication?
The materials and methods sections has a series of reagent lists (3.1 - 3.5) of what was used - this must be written into the methods sections as required. Also the strain and plasmids need to be in a table format.
For the results sections, just write the results - so th following (starting with line 158) is not results? "We expected that once the biomass reached high-density values, we induced IPTG expression, and a large number of cells would begin to produce the rec5SE-33-A2P protein, which would be cleaved into SE-33 peptides during translation. Since some time was 1required for interaction with the membrane, it was expected to occur on the cytosolic side, and the activity of the derived peptide was not supposed to be high, we expected that cells would have time to produce some amount of the SE-33-A2P peptide before the start of cell death.
Figure 3? What are the SE or SD of the experiment here - what was n?
Table 1 and figure 3 are both the same data get rid of one.
Keep figure 4 (What are the SE or SD of the experiment here - what was n?), get rid of table 2 - handle table 2 material in the methods and materials
OE is not a common acronym - do you mean optical density? OD?
Line 199. What did the studies in reference 51 state? Based on the studies [51], it was decided to use cholesterol to stabilize the membrane.
Same as above for Fig 5 and table 3, one or the other - plus any statistics that might have been used??
Figure 6 has almost no explanation and the text provided is not enough what are 1 through 6 in the figure? and you cannot see the descriptions of the X and Y axis?
Lines 220 through 223 require more explanation.
What the EC100?? - why cannot you not describe the MICs and MBCs completed with CLSI instructions?
Lines 261 to 266 describe something but it is so weird to test antimicrobials in such a way? Just MICS and MBCs?
The rest is overwritten and needs to be rewritten concisely.
Comments on the Quality of English LanguageIt is fine just horribly overwritten.
Author Response
Dear Reviewer! Thank you very for your review. We have responded to all the comments, corrections have been made. We would like to draw your attention to the fact that due to the presence of comments from other experts, the revised text also contains additional edits.

Round 2
Reviewer 3 Report
Comments and Suggestions for Authors
Rewrite sentences 9-11. The SE-33 peptide consists of THE A2P and SE-33 peptides and is a analog of cathelicidin possessing antimicrobial activity against both Gram-positive and Gram-negative bacteria, and A2P is a self-cleaving peptide.
Rewrite sentences 120 - 125 - We have developed a method for the isolation of the SE-33-A2P peptide from prokaryotic cells which we characterized by physicochemical methods and susceptibility testing. This isolation method makes it possible to obtain SE-33-A2P AMP at a relatively low cost.
Lines 179-182. These are not "buffers", I presume they are just LB with different carbon and energy sources added? Also the suthors are not looking at expression of their vector construct they are looking at OD changes????
"The use of LB buffers with different concentrations of glucose and glycerol showed that spontaneous expression disappears completely when using Buffer 3 (0.2% glucose and 0.15% glycerol), as can be seen in Figure 4. Therefore, further cultivation was carried out in the LB buffer containing 0.2% glucose and 0.15% glycerol in addition to kanamycin"
OD throughout the manuscript should read "OD560nm = 3.0" or something similar.
Line 196. "Based on the culture studies during cultivation [65], the optimal concentration of the cell culture for induction was selected". What was the optimal concentration?
Line 228. ......mass (REMOVE THE) spectrometry and HPLC
Remove Line 284 "An average MBC was 45 μg/ml for Gram-negative bacteria and 35 μg/ml for Gram-positive bacteria (Table 1)" just state the MBCs.
Authors are looking at total cell death - but not really following any standard MBC protocol I am aware of - is there are reference for their so-called MBC protocol??
Why did the authors use mouse excreta to test their peptides, is there literature that supports this approach.
Conclusions should be 5- 10 sentences - what is written know is too long.
Comments on the Quality of English Language
an English editor may be helpful
Author Response
Dear reviewer, thank you very much for you substantial evaluation of our manuscript. The answers are given in the attached file.
